# An integrated hospital-district performance evaluation for communicable diseases in low- and middle-income countries: Evidence from a pilot in three sub-Saharan countries

Lara Tavoschi[1⊙], Paolo Belardi[2⊙]*, Sara Mazzilli[3], Fabio Manenti[4], Giampietro Pellizzer[4], Desalegn Abebe[5], Gaetano Azzimonti[6], John Bosco Nsubuga[7], Giovanni Dall'Oglio[8], Milena Vainieri[2]

1 Department of Translational Research and New Technologies in Medicine and Surgery, University of Pisa, Pisa, Italy, 2 Management and Healthcare Laboratory, Institute of Management and Department EMbeDS, Sant'Anna School of Advanced Studies, Pisa, Italy, 3 Scuola Normale Superiore, Pisa, Italy, 4 Doctors with Africa CUAMM, Padova, Italy, 5 Doctors with Africa CUAMM, St. Luke Wolisso Hospital/Wolisso Catchment Area, Wolisso, Ethiopia, 6 Doctors with Africa CUAMM, Tosamaganga District Designated Hospital/Iringa District Council, Iringa, Tanzania, 7 Doctors with Africa CUAMM, St. Kizito Matany Hospital/Napak District, Matany, Uganda, 8 Doctors with Africa CUAMM, Pope John XXIII Aber Hospital/Oyam District, Gulu, Uganda

⊙ These authors contributed equally to this work.
* p.belardi@santannapisa.it

**Data Availability Statement:** At health district level, we used publicly available data that were extracted from the District Health Information

## Abstract

### Introduction

The last two decades saw an extensive effort to design, develop and implement integrated and multidimensional healthcare evaluation systems in high-income countries. However, in low- and middle-income countries, few experiences of such systems implementation have been reported in the scientific literature. We developed and piloted an innovative evaluation tool to assess the performance of health services provision for communicable diseases in three sub-Saharan African countries.

### Material and methods

A total of 42 indicators, 14 per each communicable disease care pathway, were developed. A sub-set of 23 indicators was included in the evaluation process. The communicable diseases care pathways were developed for Tuberculosis, Gastroenteritis, and HIV/AIDS, including indicators grouped in four care phases: prevention (or screening), diagnosis, treatment, and outcome. All indicators were calculated for the period 2017–2019, while performance evaluation was performed for the year 2019. The analysis involved four health districts and their relative hospitals in Ethiopia, Tanzania, and Uganda.

### Results

Substantial variability was observed over time and across the four different districts. In the Tuberculosis pathway, the majority of indicators scored below the standards and below-average performance was mainly reported for prevention and diagnosis phases. Along the

Systems (DHISs) of the countries involved in the study. At hospital level, data were extracted from each hospital's registers. All relevant data are within the paper and its Supporting information files. More particularly, data on all indicators listed in Table 1 are included in the report "Performance Evaluation System of hospital and health districts in Ethiopia, Uganda and Tanzania", that is available at the link report in Supporting information (S3). Additionally, in Supporting information (S4) we provided all data elements related to the indicators listed in Table 2.

**Funding:** This study was conducted within a research collaboration project funded by the Management and Health Laboratory of the Institute of Management of the Sant'Anna School of Advanced Studies and the NGO Doctors with Africa CUAMM. Both entities jointly worked on the study design, data collection and analysis, decision to publish, or preparation of the manuscript.

**Competing interests:** The authors have declared that no competing interests exist.

Gastroenteritis pathway, excellent performance was instead evaluated for most indicators and the highest scores were reported in prevention and treatment phases. The HIV/AIDS pathway indicators related to screening and outcome phases were below the average score, while good or excellent performance was registered within the treatment phase.

## Conclusions

The bottom-up approach and stakeholders' engagement increased local ownership of the process and the likelihood that findings will inform health services performance and quality of care. Despite the intrinsic limitations of data sources, this framework may contribute to promoting good governance, performance evaluation, outcomes measurement and accountability in settings characterised by multiple healthcare service providers.

## Introduction

Despite steady global improvements, the burden of disease due to communicable diseases (CDs) remains high in certain regions of the world, including sub-Saharan Africa. Lower respiratory infections, diarrhoeal diseases and sexually transmitted infections (including HIV) still rank high among the causes of morbidity and death [1]. Dedicated public health strategies have been designed and rolled out over the decades to tackle CDs in sub-Saharan Africa, often engaging multiple partners for their implementation. Global initiatives such as the Global Fund to Fight AIDS, Tuberculosis (TB) and Malaria and Stop-TB, have greatly accelerated progress by fostering multi-stakeholders' collaborations and resources mobilization [2]. Healthcare provision in certain areas of sub-Saharan Africa relies also on the presence of external agencies and entities such as non-governmental organizations (NGOs) [3, 4].

The heterogeneity of the healthcare services (HS) scenario poses additional challenges for performance assessment, especially in countries where national monitoring systems are suboptimal. Performance evaluation of healthcare provision is extremely relevant to i) assess the impact of different stakeholders' efforts and contributions towards the achievement of health goals; ii) monitor activities and plan policy-related initiatives; iii) set and promote quality improvement actions at national/local level; and iv) ensure accountability towards national health institutions and donors [5, 6].

In recent years, an extensive effort to design, develop and implement Performance Evaluation Systems (PESs) in healthcare have been made by institutions in single high-income countries as well as by international agencies [6–8]. The added value of PES rests on the integrated approach, taking into consideration multiple dimensions and indicators related to efficiency, structure, process, quality of care, appropriateness, and equity [9] as well as the interests of several stakeholders in the healthcare system, from a population-based perspective [10]. Yet, in low- and middle-income countries (LMICs), few experiences of PESs implementation have been reported in the scientific literature [11, 12]. Shortage of human and technical resources, unavailability and unreliability of data, lack of culture of data-driven decision making are all coexisting factors that hamper the set-up and the maintenance of coherent and integrated PES in LMICs [13–15]. Additionally, a larger body of evidence relates to the application of monitoring and evaluation frameworks of global health strategies, which usually have a vertical approach and are geared towards the assessment of progresses towards global goals rather than towards the assessment of performance [16, 17]. In this scenario, quality of health data on

CDs, as compared to other dimensions or health conditions, may have benefitted from more intense and coordinated efforts along the past decades, targeting LMICs and the sub-Saharan region in particular.

The present study aimed at investigating the feasibility of using a bottom-up and integrated PES, and at assessing the performance of health services provision related to CDs (PES-CDs) in three selected sub-Saharan countries. For this purpose, we adapted and tailored the framework developed to assess Italian regional healthcare systems performance through an iterative multi-stakeholders process [9], engaging local HSs, academia, and an Italian-based non-governmental organization (NGO) delivering health care in the target countries.

## Material and methods

### Country/Setting brief description: Similarities and differences

The analysis involved the following four hospitals and their relative health districts: i) St.Luke—Wolisso Hospital (Wolisso) and five "Woredas" in the region of Oromia in Ethiopia (Wolisso Area); ii) Tosamaganga District Designated Hospital (Tosamaganga) and the Iringa District Council (Iringa District) in the region of Iringa in Tanzania; iii) St. Kizito—Matany Hospital (Matany) and the Napak District and iv) Pope John XXIII—Aber Hospital (Aber) and the Oyam District in the northern region in Uganda, as shown in Fig 1.

The details of the analysed hospitals and their relative health districts are included in the supporting information (S1 Table).

These settings differ not only in terms of population served, the surface area covered, and, consequently, population density, but also with respect to some relevant and intrinsic factors, which include different environmental characteristics, epidemiological priorities and issues, organizational and governance models, levels of development of transport, energy, and ICT infrastructures.

Despite such differences, they were selected because of some similar characteristics inherent to all four contexts, which are related to: i) the organization of healthcare delivery across levels of care; ii) the institutional setting of the hospitals, i.e. private, faith-based and not for profit; and iii) the funding model through which hospitals are financed. Most importantly, in all these contexts clinical and administrative activities are supported by the same NGO, Doctors with Africa CUAMM (CUAMM) which contributed to strengthening health systems in LMICs through the allocation of financial and human resources, including expatriate professionals, for several decades.

### Data sources and analyses

We sourced health-related data on TB, Gastroenteritis and HIV/AIDS for the years 2017–2019. Hospital level metrics were derived from health and administrative registers, whereas district level metrics were extracted from the District Health Information Systems (DHISs) in each context of interest. In all four hospitals, TB, Gastroenteritis, and HIV/AIDS related-data were retrieved from digital and paper-based administrative department registers, which include laboratory, pharmacy, outpatient (e.g. ART clinic, TB clinic), and inpatient departments records. At health district level, data were electronically extracted from specific DHIS registers provided by the Ministry of Health of each country involved in this study. For example, TB data related to the two Ugandan sites were sourced from the Ugandan DHIS in the register 106a:3.1. Data sources are further detailed in Table 1.

Data extraction was carried out by local staff in each study context. Disease-specific datasets were created by extracting aggregated data on pre-defined variables (Table 1) from existing databases into excel spreadsheets and subsequently elaborated. Data were aggregated per

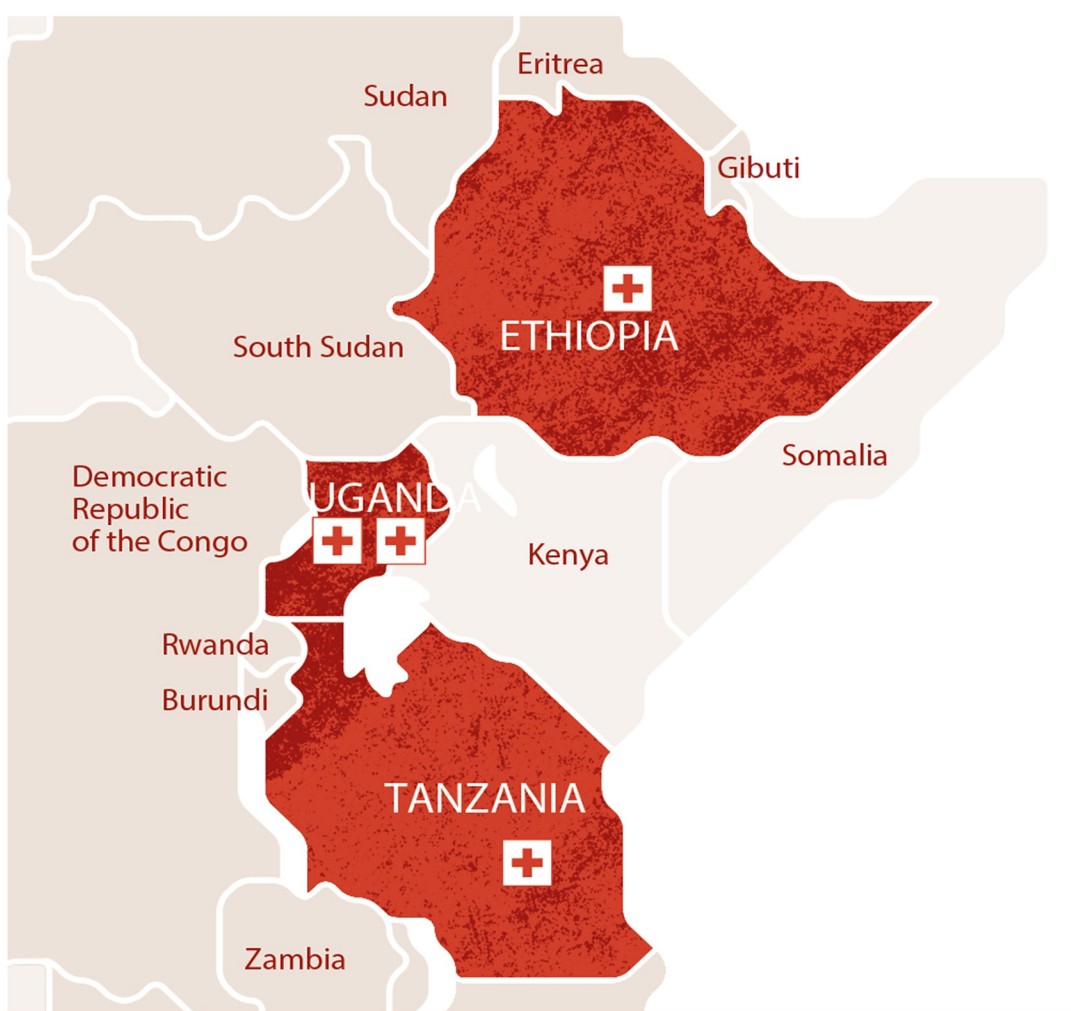

| Country | Health District | Reference Hospital | TB incidence rate at national level, per 100 000 population (1) | Percentage of deaths caused by diarrhoea in children under 5 years of age at national level (2) | Prevalence of HIV at district level (% of population aged 15-49) (3) |
|---|---|---|---|---|---|
| Ethiopia | Five «Woredas» in the Shoa-west zone | St. Luke – Wolisso Hospital | 140 | 8% | 1% |
| Tanzania | Iringa District Council | Tosamanga District Designated Hospital | 237 | 8% | 9.5% |
| Uganda | Napak District | St. Kizito – Matany Hosptital | 200 | 7% | 5.2% |
| Uganda | Oyam District | Pope John XXIII – Aber Hospital | 200 | 7% | 7% |

*Sources: (1) World Health Organization, year 2019; (2) Unicef, year 2017; (3) World Bank, year 2019*

**Fig 1. Analysed settings.** List of the four analysed hospitals and their respective health districts. Reprinted from Doctors with Africa CUAMM under a CC BY license, with permission from PLOS ONE, original copyright 2021.

**Table 1. List of the indicators included into the CDs care pathways: Tuberculosis, gastroenteritis and HIV/AIDS.**

| Phase | Indicator code | Indicator name | Numerator | Denominator | Computation level | Type of indicator | Sources |
|---|---|---|---|---|---|---|---|
| List of the indicators related to the Tuberculosis Pathway. | | | | | | | |
| Prevention | IDPT01 | Percentage of treatments with isoniazide (IPT) | Number of treatments with isoniazide (IPT) (x100) | Total number of eligible treatments | Health District | Evaluation | Tanzanian ETL/DHIS2, Ugandan eHMIS/DHIS2 (electronic sources) |
| | IDPT02 | Percentage of TB cases undergoing the HIV screening | Number of TB cases undergoing the HIV screening (x100) | Number of TB diagnosed patients | Health District | Evaluation | Ethiopian HMIS/DHIS2, Tanzanian ETL/DHIS2, Ugandan eHMIS/DHIS2 (electronic sources) |
| Diagnosis | IDPT03 | Percentage of positive TB cases on number of tests | Number of positive TB cases (confirmed by lab tests or Xpert) (x100) | Number of tests (presumptive cases) | Health District | Observation | Tanzanian ETL/DHIS2, Ugandan eHMIS/DHIS2 (electronic sources) and WHO global tuberculosis reports |
| | IDPT04 | Percentage of confirmed TB cases on diagnosed cases | Number of positive PTB cases (bacteriologically confirmed) (x100) | Number of TB diagnosed patients | Health District | Evaluation | Hospitals registers—laboratory departments (electronic and paper-based sources) |
| | IDPT05 | Percentage of confirmed PTB | Number of positive PTB cases (bacteriologically confirmed) (x100) | Number of PTB cases | Health District | Evaluation | Ethiopian HMIS/DHIS2, Tanzanian ETL/DHIS2, Ugandan eHMIS/DHIS2 (electronic sources) |
| | IDPT06 | Percentage of positive Xpert cases | Numer of positive Xpert cases (x100) | Number of Xpert cases | Hospital | Evaluation | Hospitals registers—laboratory departments (electronic and paper-based sources) |
| | IDPT06.1 | Percentage of positive Xpert RR | Number of positive Xpert RR (x100) | Number of positive Xpert | Hospital | Observation | Hospitals registers—laboratory departments (electronic and paper-based sources) |
| Treatment | IDPT07 | Percentage of treatments for extrapolmunary TB | Number of treatments "initiated" for extrapolmunary TB (x100) | Number of TB diagnoses | Health District | Evaluation | Ethiopian HMIS/DHIS2, Tanzanian ETL/DHIS2, Ugandan eHMIS/DHIS2 (electronic sources) |

(*Continued*)

**Table 1.** (Continued)

| Phase | Indicator code | Indicator name | Numerator | Denominator | Computation level | Type of indicator | Sources |
|---|---|---|---|---|---|---|---|
| Outcome | IDPT08 | Percentage of PTB MDR initiated treatments | Number of MDR initiated treatments (x100) | Number of MDR TB diagnoses | Hospital | Observation | Hospitals registers—laboratory departments (electronic and paper-based sources) |
| | IDPT09 | Percentage of cured patients | Number of cured patients (x100) | Number of PTB+ (bacteriologically confirmed) | Health District | Evaluation | Ethiopian HMIS/DHIS2, Tanzanian ETL/DHIS2, Ugandan eHMIS/DHIS2 (electronic sources) |
| | IDPT10 | Percentage of completed treatments | Number of completed treatments (x100) | Number of treated cases | Health District | Evaluation | Ethiopian HMIS/DHIS2, Tanzanian ETL/DHIS2, Ugandan eHMIS/DHIS2 (electronic sources) |
| | IDPT11 | Percentage of deaths | Number of deaths (x100) | Number of treated cases | Health District | Observation | Ethiopian HMIS/DHIS2, Tanzanian ETL/DHIS2, Ugandan eHMIS/DHIS2 (electronic sources) |
| | IDPT12 | Percentage of treatment interrupted | Number of treatments interrupted (x100) | Number of treated cases | Health District | Evaluation | Ethiopian HMIS/DHIS2, Tanzanian ETL/DHIS2, Ugandan eHMIS/DHIS2 (electronic sources) |
| | IDPT13 | Percentage of admitted patients due to TB | Number of admitted patients for TB in reference hospital (x100) | Total number of TB cases at residence level | Health District | Observation | Hospitals registers—medical departments (electronic sources) and Ethiopian HMIS/DHIS2, Tanzanian ETL/DHIS2, Ugandan eHMIS/DHIS2 (electronic sources) |

List of the indicators related to the Gastroenteritis Pathway.

| Phase | Indicator code | Indicator name | Numerator | Denominator | Computation level | Type of indicator | Sources |
|---|---|---|---|---|---|---|---|
| Prevention | B7.9 | Vaccination coverage for rota virus | Number of children under one year of age who have received 2nd dose of Rotavirus vaccine (x100) | Number of infants aged less than 1 year | Health District | Evaluation | Ethiopian HMIS/DHIS2, Tanzanian DHIS2, Ugandan eHMIS/DHIS2 (electronic source) |
| | IDPD02 | Average number of water sources by Hospital | Number of water taps | Total wards and outpatient rooms | Hospital | Evaluation | Hospital technical departments |
| | IDPD03 | Availability of an hand washing programme (Hospital) | - | - | Hospital | Observation | Hospital technical departments |
| | IDPD04 | Average number of toilets per beds in IPD | Number of toilets | Number of beds | Hospital | Evaluation | Hospital technical departments |
| | IDPD05 | Average number of toilets in OPD per number of rooms | Number of toilets in OPD | Number of rooms in OPD | Hospital | Evaluation | Hospital technical departments |

(*Continued*)

**Table 1.** (Continued)

| Phase | Indicator code | Indicator name | Numerator | Denominator | Computation level | Type of indicator | Sources |
|---|---|---|---|---|---|---|---|
| Diagnosis | IDPD06 | Percentage of positive stool tests (for parasites) | Number of positive stool tests (for parasites) (x100) | Total faeces examinations | Hospital | Observation | Hospitals registers—laboratory departments (electronic and paper-based sources) |
| | IDPD07 | Percentage of gastroenteritis diagnosed (<5 years—Outpatient) | Number of gastroenteritis diagnosed (<5 years) in OPD and HCs (x100) | Number of OPD access for children <5yr | Health District | Observation | Tanzanian DHIS2, Ugandan eHMIS/DHIS2 (electronic sources) |
| | IDPD08 | Percentage of gastroenteritis diagnosed (>5 years—Outpatient) | Number of gastroenteritis diagnosed (>5 years) in OPD and HCs (x100) | Number of OPD access >5yr | Health District | Observation | Tanzanian DHIS2, Ugandan eHMIS/DHIS2 (electronic sources) |
| | IDPD09 | Percentage of diarrhoea cases with severe dehydration due to gastroenteritis and diarrhoea | Number of diarrhoea cases with severe dehydration (x100) | Total number of cases | Hospital | Observation | Wolisso and Matany hospital's registers, Tanzanian DHIS2, Ugandan eHMIS/DHIS2 (electronic sources) |
| | IDPD10 | Percentage of discharged patients for diarrhoea and gastroenteritis | Number of discharged patients for diarrhoea and gastroenteritis (x100) | Total number of discharged patients (adults and children) | Hospital | Evaluation | Wolisso and Matany hospital's registers, Tanzanian DHIS2, Ugandan eHMIS/DHIS2 (electronic sources) |
| | IDPD11 | Percentage of diarrhoea cases (<1 year) | Number of diarrhoea cases (<1 year—acute cases) (x100) | Total number of diarrhoea cases | Health District | Observation | Tanzanian DHIS2, Ugandan eHMIS/DHIS2 (electronic sources) |
| Treatment | IDPD12 | Average number of ORS packages delivered per patient with diarrhoea (<5years) | Number of ORS packages delivered (Hospital + HCs) | Total number of diarrhoea cases (<5 years) | Health District | Evaluation | Ugandan eHMIS/DHIS2 (electronic sources) |
| | IDPD13 | Average number of Zinc Tablets doses delivered per patient with diarrhoea (<5years) | Number of Zinc Tablets doses delivered (Hospital + HCs) | Total number of diarrhoea cases (<5 years) | Health District | Evaluation | Ugandan eHMIS/DHIS2 (electronic sources) |
| Outcome | IDPD14 | Percentage of deaths with a diagnosis of gastroenteritis | Number of deaths diagnosed with gastroenteritis (patients aged < 5 years) (x100) | Number of discharged patients with a diagnosis of gastroenteritis (patients aged < 5 years) | Hospital | Evaluation | Wolisso and Matany hospitals registers, Tanzanian DHIS2, Ugandan eHMIS/DHIS2 (electronic sources) |
| | IDPD15 | ALOS for gastroenteritis | Number of inpatient days for gastroenteritis | Total number of inpatients (for gastroenteritis) | Hospital | Observation | Wolisso and Matany hospitals registers—medical department (electronic sources) |

List of the indicators related to the HIV/AIDS Pathway.

(*Continued*)

**Table 1.** (Continued)

| Phase | Indicator code | Indicator name | Numerator | Denominator | Computation level | Type of indicator | Sources |
|---|---|---|---|---|---|---|---|
| Screening | CPHIV01 | Percentage of HIV screening coverage | Number of performed tests (x100) | Number of admissions in OPD (hospital and HCs) and IPD | Health District | Observation | Ethiopian HMIS/DHIS2, Tanzanian ETL/DHIS2, Ugandan eHMIS/DHIS2 (electronic sources) |
| | CPHIV02 | Percentage of performed tests to pregnant women | Number of HIV performed tests to pregnant women followed in RCH (x100) | Total number of pregnant women with at least one ANC visit | Health District | Evaluation | Ethiopian HMIS/DHIS2, Tanzanian ETL/DHIS2, Ugandan eHMIS/DHIS2 (electronic sources) |
| | IDPT02 | Percentage of TB cases undergoing the HIV screening | Number of TB cases undergoing the HIV screening (x100) | Number of TB diagnosed patients | Health District | Evaluation | Ethiopian HMIS/DHIS2, Tanzanian ETL/DHIS2, Ugandan eHMIS/DHIS2 (electronic sources) |
| | CPHIV03 | Percentage of HIV + cases undergoing the TB screening | Number of HIV cases undergoing the TB screening (sputum, symptom questionnaire, Xpert) (x100) | Number of HIV+ cases | Health District | Evaluation | Ethiopian HMIS/DHIS2, Tanzanian ETL/DHIS2, Ugandan eHMIS/DHIS2 (electronic sources) |
| | CPHIV03.1 | Percentage of HIV patients screened for TB with Xpert | Number of HIV patients screened with Xpert for TB (x100) | Number of HIV + screened patients for TB | Hospital | Observation | Hospitals registers—laboratory departments (electronic and paper-based sources) and Ethiopian HMIS/DHIS2, Tanzanian ETL/DHIS2, Ugandan eHMIS/DHIS2 (electronic sources) |
| Diagnosis | CPHIV04 | Percentage of new diagnosed patients with CD4 < 350cell/ml | Number of diagnosed patients with CD4 < 350cell/ml (x100) | Number of new diagnosed HIV + patients | Hospital | Observation | Hospitals registers—laboratory departments (electronic and paper-based sources) and Ugandan eHMIS/DHIS2 (electronic sources) |
| | CPHIV05 | Percentage of HIV + patients with opportunistic infections (or advanced HIV) | Number of HIV+ patients with opportunistic infections diagnosed at the time of HIV diagnosis (x100) | Number of new HIV + patients diagnosed | Hospital | Observation | Hospitals registers—laboratory departments (electronic and paper-based sources) and Ugandan eHMIS/DHIS2 (electronic sources) |
| | CPHIV06 | Percentage of malnourished patients followed in a HIV unit | Number of HIV + malnourished patients currently on ART in a HIV unit (x100) | Number of patients currently in HIV unit | Health District | Observation | Ethiopian HMIS/DHIS2, Tanzanian ETL/DHIS2, Ugandan eHMIS/DHIS2 (electronic sources) |

(*Continued*)

**Table 1.** (Continued)

| Phase | Indicator code | Indicator name | Numerator | Denominator | Computation level | Type of indicator | Sources |
|---|---|---|---|---|---|---|---|
| Treatment | CPHIV07 | Percentage of new HIV + linked to ART | Number of HIV+ starting ART (x100) | Number of new patients tested HIV+ in OPD and IPD | Health District | Evaluation | Ethiopian HMIS/DHIS2, Tanzanian ETL/DHIS2, Ugandan eHMIS/DHIS2 (electronic sources) |
| | CPHIV08 | Coverage rate of the therapy | Number of HIV+ patients currently on ART therapy (x100) | Number of HIV + residents (estimated) | Health District | Evaluation | Ethiopian HMIS/DHIS2, Tanzanian ETL/DHIS2, Ugandan eHMIS/DHIS2 (electronic sources) |
| | CPHIV09 | Average number of nutritional supplements delivered per patients currently on ART therapy | Number of nutritional supplements (Plumpinat, enriched flawour ect.) delivered | Number of patients currently on ART therapy | Health District | Observation | Ugandan eHMIS/DHIS2 (electronic sources) |
| | CPHIV10 | Percentage of VL tests over the patient undergoing ART therapy | Number of patients undergoing VL tests (x100) | Number of patients currently on ART therapy | Health District | Evaluation | Hospitals registers—ART clinic/CDC departments (paper-based sources) and Tanzanian ETL/DHIS2, Ugandan eHMIS/DHIS2 (electronic sources) |
| Outcome | CPHIV11 | Percentage of patients undergoing ART therapy and tested with VL with suppression of viremia | Number of patients undergoing VL tests with viremia suppression (x100) | Number of patients currently on ART therapy and tested with VL within last 12 months | Hospital | Evaluation | Hospitals registers—ART clinic/CDC departments (paper-based sources) and Tanzanian ETL/DHIS2, Ugandan eHMIS/DHIS2 (electronic sources) |
| | CPHIV12 | Percentage of deaths undergoing ART therapy (within 12 months) | Number of patients undergoing ART therapy who died within 12 months from the beginning of the therapy (x100) | Number of patients who started ART therapy as of at least 12 months | Hospital | Observation | Ethiopian HMIS/DHIS2, Tanzanian ETL/DHIS2, Ugandan eHMIS/DHIS2 (electronic sources) |
| | CPHIV13 | ALOS (HIV admitted patients) | Number of inpatient days for HIV and its complication | Number of inpatients for HIV and its complications | Hospital | Observation | Wolisso and Matany hospitals registers—medical department (electronic sources) |

variable of interest and per year of study. Disease-specific datasets were successively shared with Italian partners for analyses validation purposes. Data analyses and relative visual representations were run using SAS (Statistical Analysis System) version 9.4.

## CDs care pathways development

This pilot study represents the result of constructive research carried out by two Italian academic research centres on healthcare economics and management, public health in collaboration with CUAMM. The study protocol/research approach used has been previously described elsewhere [Belardi et al., under review].

A multidisciplinary team elaborated the list of indicators populating the PES-CDs and a total number of 42 indicators, 14 for each specific CD care pathway, were selected. Additionally, 20 indicators were calculated at hospital level and 22 at health district level (Table 1).

Indicators were derived from international reference sources, including WHO relevant global monitoring frameworks [18–23]. Reference indicators were revised and adapted based on local needs or to data availability at hospital and health district levels. The final list of indicators with reference sources is presented in Table 1.

All the indicators were calculated for the years 2017–2019. For 16 indicators (three for TB, seven for Gastroenteritis, six for HIV/AIDS) data sources were not available for at least in one setting. Subsequently, each of the indicators calculated was assessed according to five dimensions [24], namely: relevance (is the indicator appropriate in relation to the peculiarities of the context analysed?); validity (is the indicator compliant with the purposes for which has been defined?); reliability (is the data and its sources authentic, solid, and reliable?); interpretability (does the indicator provide a univocal indication on how to interpret the data?); feasibility (can the indicator be calculated by using the existent information flows?; a reference standard is available?; is the reference standard appropriate for all contexts studied?). Using a consensus-building approach, a sub-set of 23 indicators per each disease-specific PES-CD was derived, covering all phases of care (prevention or screening, diagnosis, treatment and outcome) (Table 2). Evaluation was performed for the year 2019.

The evaluation standards were determined based on: i) international and global targets, when available; ii) on the guidelines already implemented by the Italian Inter-Regional Performance Evaluation System (IRPES) [9, 25]; or iii) on benchmarking assessment of indicators' values statistical distribution (i.e. by applying 0 to 5 scores to five bands that consider the statistical distribution of indicators values).

The indicators were evaluated using the approach already adopted for the development of the IRPES in Italy [9, 25], and subsequently applied in some OECD countries [26–28]. To provide an integrated and continuous view of performance across different settings and providers, each care pathway was represented by the music stave (the "stave") [9]. The staves illustrate strengths and weaknesses characterizing patient's pathway along the continuum of care.

## Results

Overall, the 2017–2019 trend was analysed for the 42 indicators. Substantial variability was observed over time and across the four different settings (Fig 5 and supporting information (S1 File)).

Here we present the performance results of the three staves (Figs 2–4) and one selected indicator for each stave (Fig 5).

### Stave and indicators related to the TB care pathway performance

Fig 2 shows the stave related to the TB care pathway performance evaluation for the four geographical areas considered in this study.

More particularly, in Wolisso Area, the indicators evaluating the TB prevention phase scored on the orange band, while the ones belonging to the diagnosis phase from red to yellow. Average performance was also observed for the indicator evaluating the treatment phase, while the outcome phase included indicators with scores ranging from poor to excellent performance.

In Iringa District, the performance of the indicator assessing the percentage of isoniazide preventive therapy (IPT) (IDPT01) scored on the red band, while the performance of the percentage of TB patients who underwent an HIV screening (IDPT02) scored excellent. Additionally, the indicators included in the diagnosis phase scored very poor, whereas an excellent performance was registered for the indicator evaluating the treatment phase. Finally, an ups and downs trend emerged by observing the outcome indicators.

With reference to Napak District, the stave reported a very poor or poor performance for all the evaluated indicators along the four care pathway phases.

Finally, in Oyam District, the performance of the two indicators regarding the prevention phase scored very poor and poor, respectively. Higher performance scores were registered,

**Table 2. Care pathways' indicators.**

| Care pathway | Phase | Indicator code | Indicator name | Computation level | Numerator | Denominator | Sources | Standard |
|---|---|---|---|---|---|---|---|---|
| Tuberculosis | Prevention | IDPT01 | Percentage of treatments with isoniazide (IPT) | Health District | Number of treatments with isoniazide (IPT) (x100) | Total number of eligible treatments | Tanzanian ETL/DHIS2, Ugandan eHMIS/DHIS2 (electronic sources) | International standard (WHO) |
| | | IDPT02 | Percentage of TB cases undergoing the HIV screening | Health District | Number of TB cases undergoing the HIV screening (x100) | Number of TB diagnosed patients | Ethiopian HMIS/DHIS2, Tanzanian ETL/DHIS2, Ugandan eHMIS/DHIS2 (electronic sources) | International standard (WHO) |
| | Diagnosis | IDPT04 | Percentage of confirmed TB cases on diagnosed cases | Health District | Number of positive PTB cases (bacteriologically confirmed) (x100) | Number of TB diagnosed patients | Hospitals registers —laboratory departments (electronic and paper-based sources) | International standard (WHO) |
| | | IDPT05 | Percentage of confirmed PTB | Health District | Number of positive PTB cases (bacteriologically confirmed) (x100) | Number of PTB cases | Ethiopian HMIS/DHIS2, Tanzanian ETL/DHIS2, Ugandan eHMIS/DHIS2 (electronic sources) | International standard (WHO) |
| | | IDPT06 | Percentage of positive Xpert cases | Hospital | Numer of positive Xpert cases (x100) | Number of Xpert cases | Hospitals registers —laboratory departments (electronic and paper-based sources) | International standard (WHO) |
| | Treatment | IDPT07 | Percentage of treatments for extrapolmunary TB | Health District | Number of treatments "initiated" for extrapolmunary TB (x100) | Number of TB diagnoses | Ethiopian HMIS/DHIS2, Tanzanian ETL/DHIS2, Ugandan eHMIS/DHIS2 (electronic sources) | International standard (WHO) |
| | Outcome | IDPT09 | Percentage of cured patients | Health District | Number of cured patients (x100) | Number of PTB+ (bacteriologically confirmed) | Ethiopian HMIS/DHIS2, Tanzanian ETL/DHIS2, Ugandan eHMIS/DHIS2 (electronic sources) | International standard (WHO) |
| | | IDPT10 | Percentage of completed treatments | Health District | Number of completed treatments (x100) | Number of treated cases | Ethiopian HMIS/DHIS2, Tanzanian ETL/DHIS2, Ugandan eHMIS/DHIS2 (electronic sources) | International standard (WHO) |
| | | IDPT12 | Percentage of treatment interrupted | Health District | Number of treatments interrupted (x100) | Number of treated cases | Ethiopian HMIS/DHIS2, Tanzanian ETL/DHIS2, Ugandan eHMIS/DHIS2 (electronic sources) | International standard (WHO) |

(*Continued*)

**Table 2.** (Continued)

| Care pathway | Phase | Indicator code | Indicator name | Computation level | Numerator | Denominator | Sources | Standard |
|---|---|---|---|---|---|---|---|---|
| Gastroenteritis | Prevention | B7.9 | Vaccination coverage for rota virus | Health District | Number of children under one year of age who have received 2nd dose of Rotavirus vaccine (x100) | Number of infants aged less than 1 year | Ethiopian HMIS/ DHIS2, Tanzanian DHIS2, Ugandan eHMIS/DHIS2 (electronic source) | International standard (IRPES) |
| | | IDPD02 | Average number of water sources by Hospital | Hospital | Number of water taps | Total wards and outpatient rooms | Hospital technical departments | International standard (Infection Prevention Control of WHO) |
| | | IDPD04 | Average number of toilets per beds in IPD | Hospital | Number of toilets | Number of beds | Hospital technical departments | International standard (Infection Prevention Control of WHO) |
| | | IDPD05 | Average number of toilets in OPD per number of rooms | Hospital | Number of toilets in OPD | Number of rooms in OPD | Hospital technical departments | International standard (WHO) |
| | Diagnosis | IDPD10 | Percentage of discharged patients for diarrhoea and gastroenteritis | Hospital | Number of discharged patients for diarrhoea and gastroenteritis (x100) | Total number of discharged patients (adults and children) | Wolisso and Matany hospital's registers, Tanzanian DHIS2, Ugandan eHMIS/ DHIS2 (electronic sources) | Benchmarking assessment of values statistical distribution |
| | Treatment | IDPD12 | Average number of ORS packages delivered per patient with diarrhoea (<5years) | Health District | Number of ORS packages delivered (Hospital + HCs) | Total number of diarrhoea cases (<5 years) | Ugandan eHMIS/ DHIS2 (electronic sources) | International standard (WHO) |
| | | IDPD13 | Average number of Zinc Tablets doses delivered per patient with diarrhoea (<5years) | Health District | Number of Zinc Tablets doses delivered (Hospital + HCs) | Total number of diarrhoea cases (<5 years) | Ugandan eHMIS/ DHIS2 (electronic sources) | International standard (WHO) |
| | Outcome | IDPD14 | Percentage of deaths with a diagnosis of gastroenteritis | Hospital | Number of deaths diagnosed with gastroenteritis (patients aged < 5 years) (x100) | Number of discharged patients with a diagnosis of gastroenteritis (patients aged < 5 years) | Wolisso and Matany hospitals registers, Tanzanian DHIS2, Ugandan eHMIS/ DHIS2 (electronic sources) | Benchmarking assessment of values statistical distribution |

(*Continued*)

**Table 2.** (Continued)

| Care pathway | Phase | Indicator code | Indicator name | Computation level | Numerator | Denominator | Sources | Standard |
|---|---|---|---|---|---|---|---|---|
| HIV/AIDS | Screening | CPHIV02 | Percentage of performed tests to pregnant women | Health District | Number of HIV performed tests to pregnant women followed in RCH (x100) | Total number of pregnant women with at least one ANC visit | Ethiopian HMIS/DHIS2, Tanzanian ETL/DHIS2, Ugandan eHMIS/DHIS2 (electronic sources) | International standard (WHO) |
| | | IDPT02 | Percentage of TB cases undergoing the HIV screening | Health District | Number of TB cases undergoing the HIV screening (x100) | Number of TB diagnosed patients | Ethiopian HMIS/DHIS2, Tanzanian ETL/DHIS2, Ugandan eHMIS/DHIS2 (electronic sources) | International standard (WHO) |
| | | CPHIV03 | Percentage of HIV+ cases undergoing the TB screening | Health District | Number of HIV cases undergoing the TB screening (sputum, symptom questionnaire, Xpert) (x100) | Number of HIV+ cases | Ethiopian HMIS/DHIS2, Tanzanian ETL/DHIS2, Ugandan eHMIS/DHIS2 (electronic sources) | International standard (WHO) |
| | Treatment | CPHIV07 | Percentage of new HIV+ linked to ART | Health District | Number of HIV+ starting ART (x100) | Number of new patients tested HIV+ in OPD and IPD | Ethiopian HMIS/DHIS2, Tanzanian ETL/DHIS2, Ugandan eHMIS/DHIS2 (electronic sources) | International standard (WHO) |
| | | CPHIV08 | Coverage rate of the therapy | Health District | Number of HIV+ patients currently on ART therapy (x100) | Number of HIV+ residents (estimated) | Ethiopian HMIS/DHIS2, Tanzanian ETL/DHIS2, Ugandan eHMIS/DHIS2 (electronic sources) | International standard (WHO) |
| | | CPHIV10 | Percentage of VL tests over the patient undergoing ART therapy | Health District | Number of patients undergoing VL tests (x100) | Number of patients currently on ART therapy | Hospitals registers—ART clinic/CDC departments (paper-based sources) and Tanzanian ETL/DHIS2, Ugandan eHMIS/DHIS2 (electronic sources) | International standard (WHO) |
| | Outcome | CPHIV11 | Percentage of patients undergoing ART therapy and tested with VL with suppression of viremia | Hospital | Number of patients undergoing VL tests with viremia suppression (x100) | Number of patients currently on ART therapy and tested with VL within last 12 months | Hospitals registers—ART clinic/CDC departments (paper-based sources) and Tanzanian ETL/DHIS2, Ugandan eHMIS/DHIS2 (electronic sources) | International standard (WHO) |

List of the evaluated indicators included into the CDs pathways.

instead, along the diagnosis and treatment phases, with a negative deterioration in both indicators of the outcome phase, which scored on the red band.

Fig 3 reports the three-year trend of the percentage of TB patients who completed the treatment in the reference year (indicator IDPT10, outcome phase). The target of 90% was set

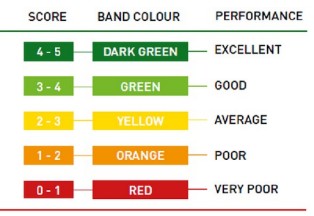

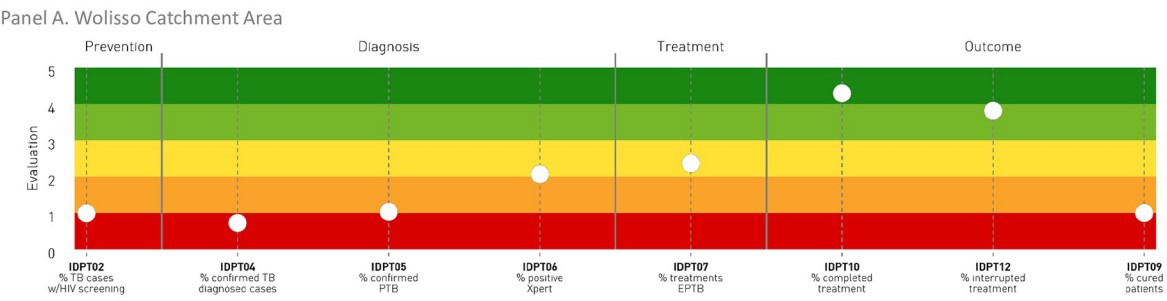

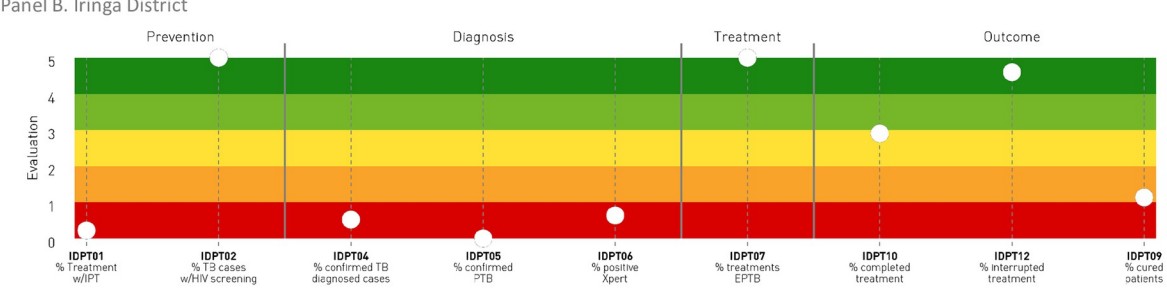

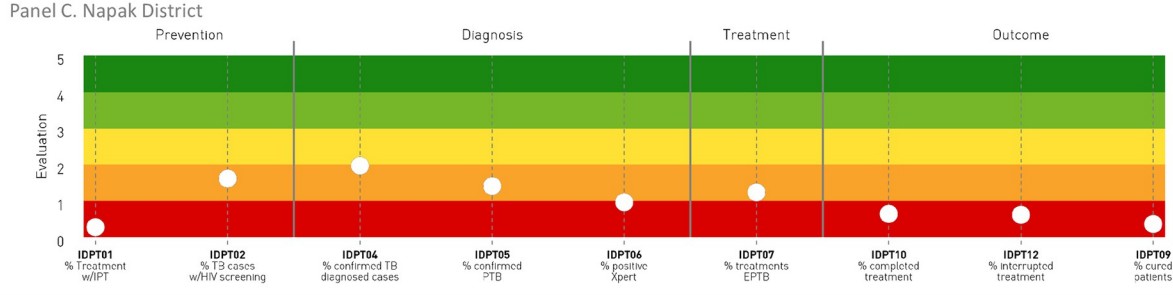

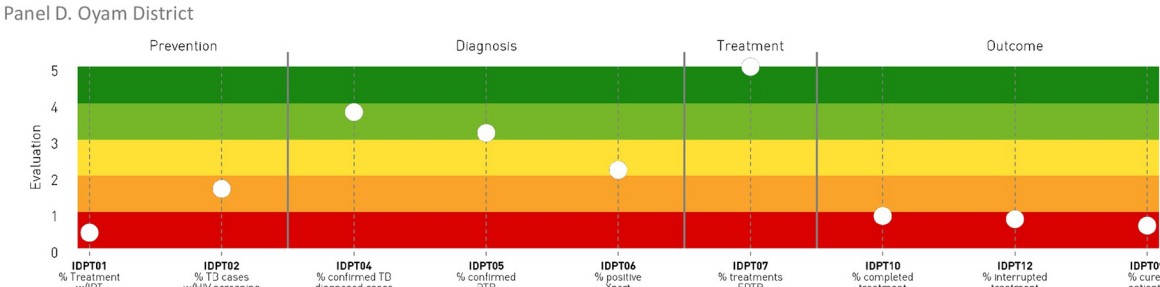

**Fig 2. Tuberculosis care pathway.** Performance of care pathways related to tuberculosis.

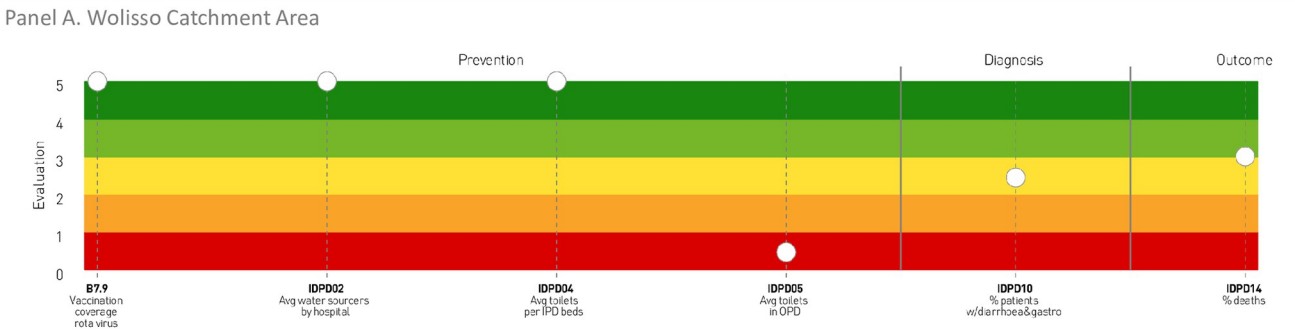

Panel A. Wolisso Catchment Area

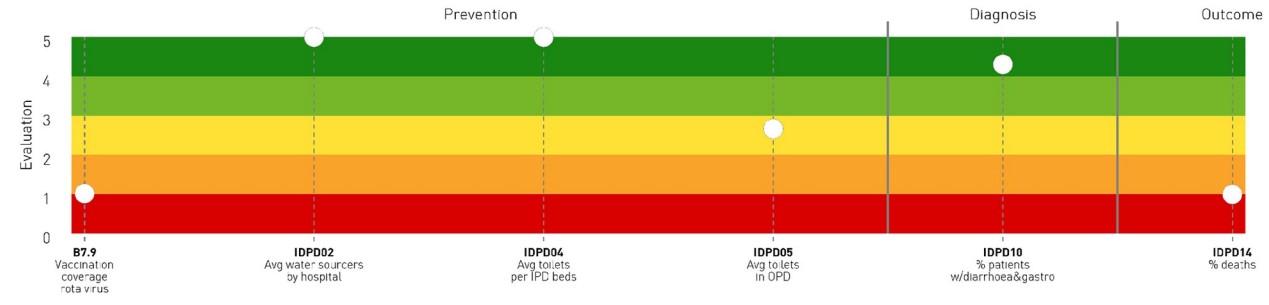

Panel B. Iringa District

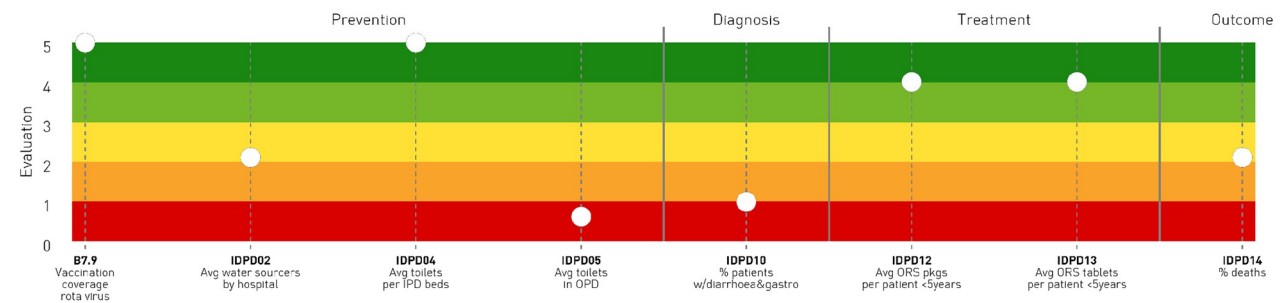

Panel C. Napak District

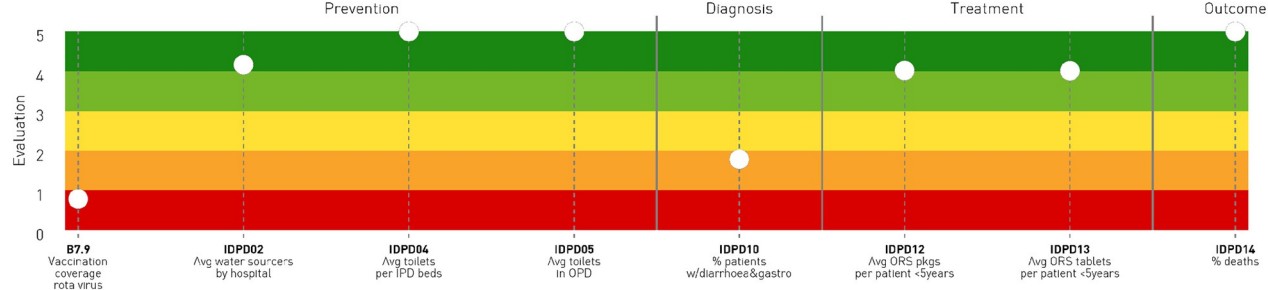

Panel D. Oyam District

**Fig 3. Gastroenteritis care pathway.** Performance of care pathways related to gastroenteritis.

Panel A. Wolisso Catchment Area

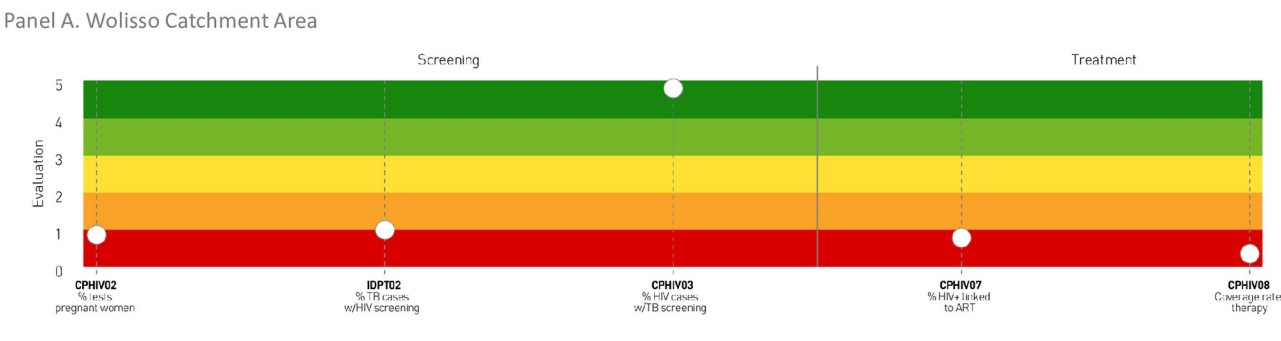

Panel B. Iringa District

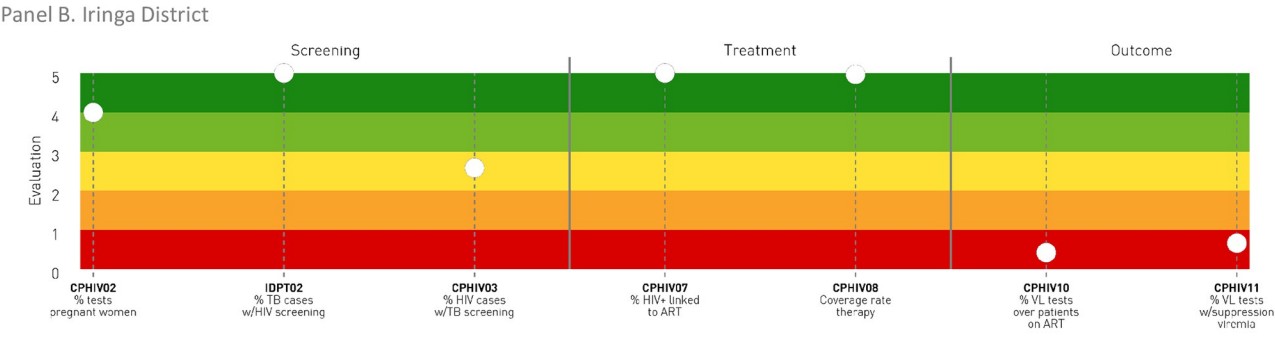

Panel C. Napak District

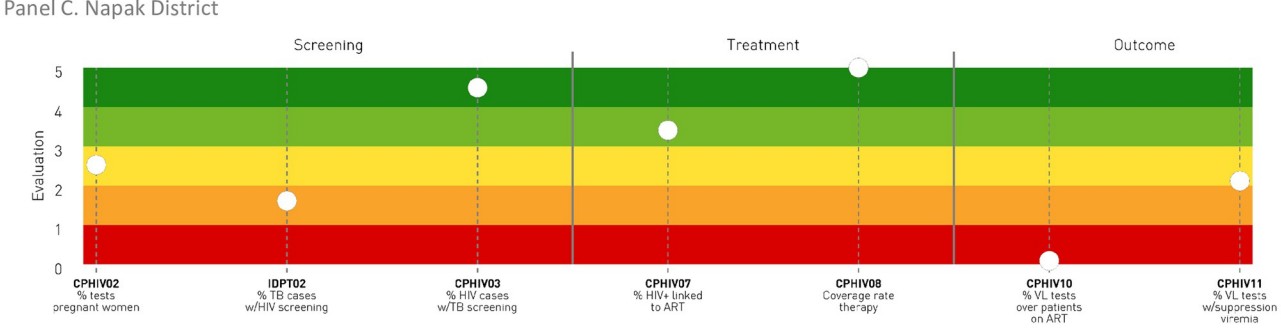

Panel D. Oyam District

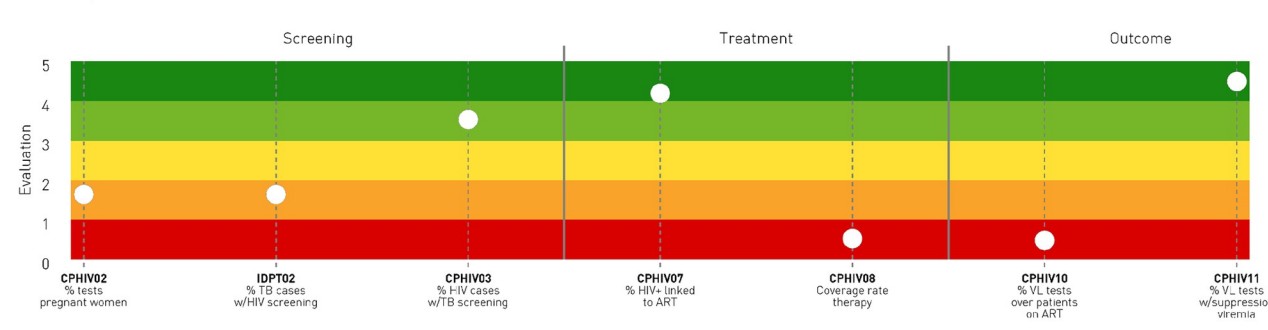

**Fig 4. HIV/AIDS care pathway.** Performance of care pathways related to HIV/AIDS.

**IDPT10 Percentage of completed tuberculosis treatments**
Computational level: District

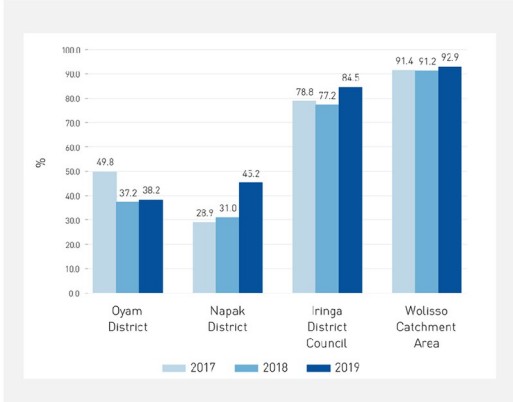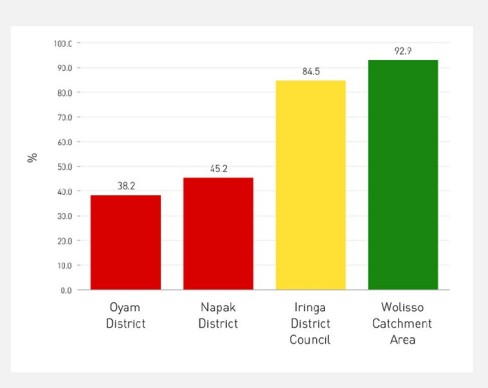

**IDPD14 Percentage of deaths with a diagnosis of gastroenteritis among patients aged less than five years**
Computational level: Hospital

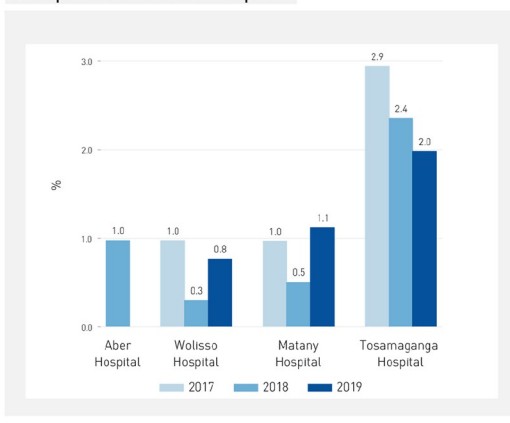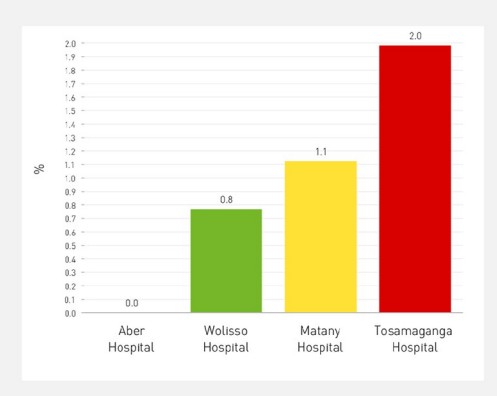

**CPHIV08 Coverage rate of the ART**
Computational level: District

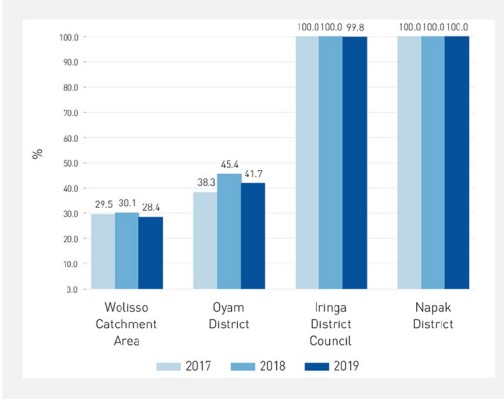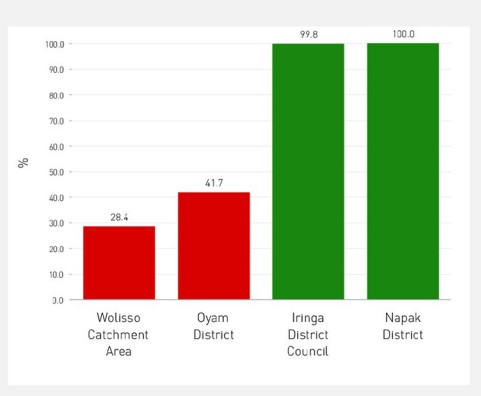

**Fig 5. One selected indicator per each stave.** Three-year trend indicators (panels on the left), and their respective evaluated scores for 2019 (panel on the right), one for each CD pathway.

based on the WHO guidelines [29]. The indicator values ranged from 38% in the Oyam District to 93% in the Wolisso Area and scored 45% and 85% in Napak and Iringa Districts respectively. For Napak District, this observation was consistent with the other indicators along the TB care pathway. The highest percentage of patients who interrupted the treatment (S1 Fig, IDPT12) and those who resulted positive to the Xpert rifampicin-resistance test (S1 Fig, IDPT06.1) were in fact reported in Napak District.

## Stave and indicators related to the gastroenteritis care pathway performance

From the analysis of the stave representing the gastroenteritis care pathway evaluation (Fig 3), it emerged that in Wolisso Area all the indicators related to the prevention phase showed excellent performance, with the exclusion of the indicator IDPD05, i.e. average number of toilets in OPD per number of rooms, which scored on the red band. Moreover, the performance of the indicators included in the diagnosis and outcome phases were registered with average and good scores, respectively. Indicators evaluating the treatment phase were not available for this setting.

With respect to Iringa District, the prevention phase showed heterogeneous results, with indicators scoring from poor to excellent performance. The indicator evaluating the diagnosis phase, i.e. the percentage of discharged patients for diarrhoea and gastroenteritis at hospital level (IDPD10), performed very well, while the one belonging to the outcome phase, namely the percentage of deaths with a diagnosis of gastroenteritis (IDPD14), scored on the orange band. Indicators evaluating the treatment phase were not available for this setting.

As in the previous stave, in Napak District the prevention phase showed diversified results in terms of performance scores, from excellent (indicators B7.9 and IDPD04) to average (IDPD02) and very poor (IDPD05). Moreover, the indicator evaluating the diagnosis phase performed poor, while higher performance scores were registered for the treatment and outcome phases.

With reference to Oyam District, the stave displayed excellent performance, except for the indicator regarding vaccination coverage for rota virus (B7.9). Additionally, the indicator related to the diagnosis phase scored poorly, whereas the indicators belonging to the other two phases showed a very good or excellent performance.

The histogram in Fig 5 shows the three-year trend of the percentage of deaths with a diagnosis of gastroenteritis and diarrhoea among hospitalized patients aged less than five years (indicator IDPD14, outcome phase). The target of 0,4% was set based on data assessment in benchmarking among the hospitals analysed. In the three-year period examined a fluctuating trend emerged. Except for Tosamaganga, the indicators values were homogeneous among districts. When comparing this indicator with the percentage of patients diagnosed with severe dehydration (S1 Fig, IDPD09), we observe some discrepancies regarding Tosamaganga. In 2019, when the highest death rate (2.0%) was reported, the hospital registered the lowest percentage of severe cases diagnosed (2.5%). Furthermore, Matany and Wolisso showed similar death rates, but widely different percentages of severe cases.

## Stave and indicators related to the HIV/AIDS care pathway performance

The stave representing the HIV/AIDS care pathway (Fig 4) did not include indicators assessing primary prevention and diagnosis activities.

In Wolisso area, the performance of all evaluated indicators belonging to the screening and treatment phases scored very poor, with the exception of the percentage of HIV positive cases

undergoing the TB screening (CPHIV03) that had an excellent score. Data on outcome measures were not available.

In Iringa District, the three indicators related to the screening phase scored average to excellent. Excellent performance results emerged from indicators belonging to the treatment phase, while those indicators evaluating treatment outcomes scored on the red band.

With respect to Napak District, the three indicators related to the screening phase scored poor to excellent, the indicators evaluating the treatment phase scored good to excellent, and those related to outcomes scored on the low and average bands.

In Oyam District, the screening phase showed heterogeneous results, with performance scores ranging from poor to good. Additionally, the two indicators evaluating the treatment phase (CPHIV07 and CPHIV08) scored on the dark green and red bands respectively, as well as those evaluating treatment outcomes (CPHIV10 and CPHIV11). Fig 5 illustrates the three-year trend of the HIV coverage rate of HIV therapy among residents at residence level (indicator CPHIV08, treatment phase). The denominator consists of the estimation of HIV prevalence among residents in the areas considered. The target of 95% was fixed according to WHO standard [30]. The graph shows that in the Napak and Oyam Districts almost all the estimation of HIV positive residents were enrolled on ART therapy over three years, while in the Wolisso Area and Oyam District this percentage was under 50%.

## Discussion

In LMICs, performance evaluation tools are often used for reporting on achievement against international targets, providing baseline assessments under specific monitoring programmes or related to definite project goals, and ensuring accountability to donors [16, 17]. Indeed, these frameworks usually imply top-down approaches intended to evaluate outcomes at macro or project level. Here, we described the pilot experience in developing and implementing a PES-CDs in three sub-Saharan African countries using a bottom-up approach.

Indeed, the PES-CDs framework was developed through a north-south cooperation involving a NGO, two research centres and local health authorities and providers. This multisectoral cooperation benefitted from the commitment of the NGO, local stakeholders and partners to strengthen health systems activities using a data-driven and evidence-based approach and represented a fundamental condition for the development of the appropriate tool. The NGO's knowledge of the institutional and epidemiological contexts allowed the design of the system rapidly and facilitated the process of interaction and integration with local communities and professionals. Local stakeholders' involvement contributed to raise awareness and foster a common understanding of the jointly conceived PES-CDs, to ensure high degree of participation during all phases of development and to engage a multidisciplinary team. Moreover, this collaboration would not have been possible without the scientific contribution of academia, whose know-how guaranteed the application of rigorous methodology to PES-CDs development and data analyses taking advantage of previous experience in other healthcare systems.

The innovative contribution of this study lies in the development of a set of indicators aimed at measuring and evaluating the performance of the patient pathway along the continuum of care within the local healthcare system. Since HSs in each geographical area involve a plurality of organizations, the design of an evaluation framework that integrates different providers and their respective perspectives becomes fundamental. The PES-CDs added value relied in the assessment of the capacity of local healthcare systems to perform along the continuum of care and through the integrations of different organizations [31]. In other words, the focus of this evaluation shifted from an organization-centred approach to a patient-focused

perspective [32], thus highlighting how local healthcare systems are capable of creating value for their reference populations [9, 10].

In our experience, the use of the stave as an intuitive and effective visual representation of the results proved to be a successful approach to highlight strengths and weaknesses along the patients' care pathway. In particular, it provided policy makers and healthcare managers with an integrated and continuous view of the performance across different healthcare settings [9].

In particular, from the TB stave analysis emerged that both TB primary (IDPT01) and secondary prevention (IDPT02) could be greatly improved in all the contexts considered, even if secondary prevention performed slightly better. When assessing outcomes, the percentage of cured patients (IDPT09) scored differently across settings, and appeared not to be related to the diagnostic or treatment performance, but rather to the rate of drop out. For example, in Oyam District, where the indicators belonging to the diagnosis and treatment phases scored high, IDPT09 scored very poor. This could be attributed to the still low treatment adherence rates due to the lack of patients awareness of the importance to complete the treatment [33]. Napak District scored poorly along the whole TB pathway, registering a low rate of treatment completion, thus highlighting the challenges in ensuring adherence to treatment in an area of predominantly nomadic population. On the contrary, Iringa and Wolisso Districts registered the highest percentages of cured TB patients suggesting a good service organization and patient's follow-up system. Appropriate and locally tailored strategies to increase treatment compliance are necessary to ensure a wider proportion of treated patients resolve the infection [34].

The Gastroenteritis care pathway was least complete due to data unavailability at both health district and hospital levels. A wide variation was observed across different settings, especially concerning the vaccination coverage for rota virus (B7.9). Indeed, such remarkable differences may be explained by the difficulty of complying with the international target (98% coverage), which is particularly challenging in these settings due to many reasons (i.e. interruption of the vaccine cold chain, natural events as well as duration of raining seasons) [35]. Information on treatment and disease outcomes was less consistent and not always available. The capacity of diagnosing and defining severe cases and their attributable deaths were not uniform across different contexts. The lack of reliability in defining causes of death is a well-known phenomenon in LMICs [36]. For example, deaths with a diagnosis of gastroenteritis and diarrhoea among hospitalized children aged <5 ranged from very poor to excellent and were not related to treatment performances. Finally, Hygiene and sanitation prevention measures met the WHO standards when implemented at the hospital level, while scored worse at the residential level, with important implications in terms of investment beyond the healthcare sector.

The HIV care pathway benefitted from a robust tradition of regional and global monitoring and vertical investments [37]. Performance indicators related to screening rarely achieved the international standards. Moreover, it was likely to have an under-reporting of HIV cases in the contexts analysed. This may have affected the estimation of treatment performance indicators and, particularly, the coverage rate of the therapy (CPHIV08), which was calculated using HIV prevalence estimates. As a consequence, the assessment of single indicators, as compared to the stave, may not provide an accurate and complete understanding of HSs provision gaps. Outcomes indicators also were affected by local practices and data availability, resulting in low accuracy and validity of the data. The most relevant example was perhaps the indicator measuring proportion of patients undergoing ART therapy and with suppression of viremia (CPHIV11) which scored from excellent to very poor across the different settings, but was likely to be severely affected by the fact that proportion of patients tested for HIV viral load during treatment scored very poorly in all the reference hospitals. This might be a very relevant

finding considering that suppression of viral load is one of the key global indicators to assess country performance against WHO goals for HIV control [38].

By applying the PES-CDs to four settings located in three different countries, we tested the tool's scalability and replicability in diversified contexts, building on previous experiences [39, 40]. This supra-national benchmarking exercise was made possible by the adoption of targets and standards defined and already in use at international level. Consequently, based on our experience, the PES-CDs indicators and graphical representation could be used by other organizations providing HS in LMICs at various levels, regardless of the epidemiological context or the organizational model.

As in high-income countries, where the HSs provision heterogeneity has been studied for over 40 years [41], in LMICs unwarranted clinical variation is also common. Our results highlighted substantial variability across all care pathway's phases and between the four settings by mean using standardised indicators and targets. The relative weight of the underlying determinants will need to be further analysed with local stakeholders.

Health professionals' motivation and engagement in PES-CDs may also vary as different monitoring activities (e.g. data capturing, collection, analysis, interpretation) are not always prioritized in the daily routine and often derive from top-down initiatives [42, 43]. Therefore, the commitment to capturing data of sufficient quality and reliability is often limited and clinicians are rarely involved in or responsible for the use of PES-CDs data for monitoring or planning improvements. In this scenario, the PES-CDs promoted data sharing between local healthcare system providers in the study settings and facilitated the development of local competencies in data collection, analysis and interpretation, including problem-solving. In addition, by using data obtained through hospitals and health districts' information systems, the PES-CDs seeks to foster synergies between existing public information sources.

For all these reasons, the main aim of the PES-CDs remains its use at local level (e.g. hospital level, district level) to boost performance evaluation over time and to promote and guide hospital/district internal quality improvement efforts.

Over and above data handling limitations already mentioned, this study constituted a pilot experience and the extent of the impact on the local settings has not been evaluated yet. Future rounds of data collection, analyses and sharing with local stakeholders are planned in the coming years with the aim of turning PES-CDs into a routine activity. Further developments of the PES-CD may also be considered to assess, among others, usefulness and impact on health system financing as well as the inclusion of additional measures to investigate the dimension of equity in healthcare access.

## Conclusion

We described an innovative approach to develop and roll-out a PES-CDs of health services provision along three CDs care pathways in three sub-Saharan African countries through multidisciplinary and north-south collaboration. Despite the intrinsic limitations of data sources, our results demonstrated that this system has great potential to strengthen the culture of data collection and monitoring at local level, ultimately fostering data-driven policy making and planning in healthcare. The bottom-up approach and stakeholders' engagement increased the likelihood of local ownership of the process and engagement to implement changes needed to improve HS performance and quality of care. In health systems characterized by multiple actors (governmental entities, private funders and providers), the proposed PES-CDs may contribute to achieve good governance in HSs provision, by stimulating performance evaluation, outcomes measurement and accountability.

## Supporting information

**S1 Fig. Trend and evaluated indicators related to the three CDs care pathways.**
(PDF)

**S1 Table. List of the analysed hospitals and their relative health districts or catchment area.**
(PDF)

**S1 File. Link to the report "Performance Evaluation System of hospital and health districts in Ethiopia, Uganda and Tanzania".**
(PDF)

**S2 File. Data elements related to the indicators listed in Table 2.**
(PDF)

## Acknowledgments

The authors would also like to thank all the professionals of the four hospitals and health districts for their precious work on data extraction and collection.

Wolisso Catchment Area: Stefano Vicentini, Stefano Parlamento, Yonas Desta, Ettore Boles. Iringa District Council: Pietro Berretta, Anza Lema, William Mbuta, Beni Tweve, Luca Brasili, Chiara Sinigaglia. Napak District: Gunther Nahrich, Apuda Daniel, Damiano Amei. Oyam District: Mattia Quargnolo, Samuel Okori, Christopher Bingom, Bobbie Okello J., Annet Ariko, Babra Muga. We also would like to thank Mauro Almaviva for his useful comments and insights.

## Author Contributions

**Conceptualization:** Lara Tavoschi, Paolo Belardi, Fabio Manenti, Giampietro Pellizzer, Milena Vainieri.

**Data curation:** Paolo Belardi, Sara Mazzilli, Desalegn Abebe, Gaetano Azzimonti, John Bosco Nsubuga, Giovanni Dall'Oglio.

**Methodology:** Lara Tavoschi, Paolo Belardi.

**Supervision:** Lara Tavoschi.

**Validation:** Lara Tavoschi, Paolo Belardi, Fabio Manenti, Giampietro Pellizzer, Desalegn Abebe, Gaetano Azzimonti, John Bosco Nsubuga, Giovanni Dall'Oglio, Milena Vainieri.

**Visualization:** Paolo Belardi.

**Writing – original draft:** Lara Tavoschi, Paolo Belardi, Sara Mazzilli.

**Writing – review & editing:** Lara Tavoschi, Paolo Belardi, Fabio Manenti, Giampietro Pellizzer, Milena Vainieri.

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
