## [Decision Letter · Decision Letter 0]

16 Dec 2021

PONE-D-21-06514An integrated hospital-district performance evaluation for communicable diseases care pathways in low-and middle-income countries: evidence from three sub-Saharan African countriesPLOS ONE

Dear Dr. Belardi,

Thank you for submitting your manuscript to PLOS ONE. After careful consideration, we feel that it has merit but does not fully meet PLOS ONE’s publication criteria as it currently stands. Therefore, we invite you to submit a revised version of the manuscript that addresses the points raised during the review process.

The reviewers raised a number of concerns about the presentation and methodological approach. Both reviewers felt that the objectives of the paper were not presented clearly, and both felt that the discussion of the results was insufficient. Their comments can be viewed in full, below.

We look forward to receiving your revised manuscript.

Kind regards,

Natasha McDonald, PhD

Associate Editor

PLOS ONE

Journal Requirements:

5. We note that Figure 1 in your submission contain map images which may be copyrighted. All PLOS content is published under the Creative Commons Attribution License (CC BY 4.0), which means that the manuscript, images, and Supporting Information files will be freely available online, and any third party is permitted to access, download, copy, distribute, and use these materials in any way, even commercially, with proper attribution. For these reasons, we cannot publish previously copyrighted maps or satellite images created using proprietary data, such as Google software (Google Maps, Street View, and Earth). For more information, see our copyright guidelines: http://journals.plos.org/plosone/s/licenses-and-copyright.

Reviewers' comments:

Reviewer's Responses to Questions

**Comments to the Author**

1. Is the manuscript technically sound, and do the data support the conclusions?

Reviewer #1: Partly

Reviewer #2: No

2. Has the statistical analysis been performed appropriately and rigorously? 

Reviewer #1: Yes

Reviewer #2: Yes

3. Have the authors made all data underlying the findings in their manuscript fully available?

Reviewer #1: Yes

Reviewer #2: Yes

4. Is the manuscript presented in an intelligible fashion and written in standard English?

Reviewer #1: Yes

Reviewer #2: No

5. Review Comments to the Author

Reviewer #1: The paper is interesting and the approach to the evaluation of health services is innovative for Sub-Saharan Africa.

However, one major general point and some minor ones should be addressed to make the manuscript valuable for pubblication.

In general the paper seems to be more focused on the methods rather than the results (more evident in the abstract and in the discussion). If you intent is to present and innovative way to evaluate health services in Africa, you could consider another type of paper, focused on methodology. Otherwise you should focus more on the results. How the services you selected performed in the TB, HIV and gastroenteritis care?

In other words the logic of the research is not linear. The objective is to evaluate the tool (PES-CDs), but the results of the research are about health services. Maybe the objective was to evaluate the health services using an innovative tool?

Minor comments:

Abstract:

- Consider structuring the abstract in sections, as you did in the main body of the paper.

- Please, include results in the abstract

Material and methods:

- is the NGO you mention CUAMM? You never state it clearly, why not?

Results:

- Line 205-206: What do you mean by “correlate”? Do you have any measure of this correlation? If not maybe a more vague wording could be more indicated.

- Line 216-219: This is more to be considered as a discussion point, not to be included in the results section.

Discussion:

- Please, start the discussion with the main findings of the paper, then start discussing them.

- Your discussion seems to be more focused on the method you used rathern than the results. However this is not a mothodological paper. The reccomandation is to rewrite the discussion starting from the main results in the performance of the services for the considered diseases and try to comapre them with similar data in the same or other contexts.

Reviewer #2: Some notes about the article:

- the present study has great potential, uses several data, a consistent instrument, but needs greater adequacy in terms of structure for exposing data, results and even discussion;

- there is a certain inconsistency between title-objective-results, making the article lengthy to read, sometimes repetitive, however, not making it clear what is, in fact, the main objective of the study;

- OBJECTIVE/Title: in my view, objective and title are not integrated and articulated, in addition to the authors mentioning different objectives throughout the text, leaving the reader confused about what is intended to be explained;

The authors mention at least 3 objectives: “evaluating the performance of the patient journey within the local healthcare system, thus assessing its capacity to provide healthcare services along the continuum of care and through the integrations of different organizations”; “we aimed at describing the development of a PES of health services provision related to CDs (PES-CDs) in three selected sub-Saharan countries”, AND “evaluating the performance of the local healthcare system along the patient continuum care”;

- the authors use the term “pathways” and “journey” giving the impression to the reader that they will deal with the paths of the patient through the health system, but in fact, I understand that these are care actions against three communicable diseases;

- at other times, the reader has the impression that the main objective of the study, in fact, is to validate an instrument for performance evaluation;

- the article has a lot of data to be explored and it might be prudent for the authors to organize a better title, objective, methods, results and discussion, so that this can be made clear to the reader;

- the present study has very robust data and is extremely important for the scientific community, as well as for the countries covered, however, it is necessary to carry out a broad review about the organization of the data and what, in fact, is intended respond with the results obtained.

6. PLOS authors have the option to publish the peer review history of their article (what does this mean?). If published, this will include your full peer review and any attached files.

Reviewer #1: No

Reviewer #2: No

---

## [Author Response · Author response to Decision Letter 0]

24 Jan 2022

Responses to each of the comments are included in the “Response to Reviewers” file.

---

## [Editor Report · Decision Letter 1]

17 Mar 2022

An integrated hospital-district performance evaluation for communicable diseases in low-and middle-income countries: evidence from a pilot in three sub-Saharan countries

PONE-D-21-06514R1

Dear Dr. Belardi,

We’re pleased to inform you that your manuscript has been judged scientifically suitable for publication and will be formally accepted for publication once it meets all outstanding technical requirements.

Kind regards,

Maria Cristina Marazzi

Guest Editor

PLOS ONE

Additional Editor Comments (optional):

Dear authors,

I participated as a reviewer for the initial evaluation of this manuscript, and it is now significantly improved according to the suggestions you received.

All the recommendations have been met.

The paper is now linear and more clear and is valuable for publication.

The study provides an interesting example of health care evaluation in sub-saharan Africa, that could contribute to knowledge and practice in public health. Hence the recommendation to accept it for publication.

Regards
---

## [Editor Report · Acceptance letter]

23 Mar 2022

PONE-D-21-06514R1 

An integrated hospital-district performance evaluation for communicable diseases in low-and middle-income countries: evidence from a pilot in three sub-Saharan countries 

Dear Dr. Belardi:

I'm pleased to inform you that your manuscript has been deemed suitable for publication in PLOS ONE. Congratulations! Your manuscript is now with our production department. 

Kind regards, 

on behalf of

Professor Maria Cristina Marazzi 

Guest Editor

PLOS ONE